# The Unfolded Protein Response in Sarcomas: From Proteostasis to Therapy Resistance

**DOI:** 10.3390/cancers17213489

**Published:** 2025-10-30

**Authors:** Elizabeta Ilieva, Sofia Avnet, Nicola Baldini, Margherita Cortini

**Affiliations:** 1Department of Biomedical and Neuromotor Sciences, Alma Mater Studiorum, Università di Bologna, 40138 Bologna, Italy; elizabeta.ilieva2@unibo.it (E.I.); sofia.avnet3@unibo.it (S.A.); nicola.baldini@ior.it (N.B.); 2Biomedical Science, Technology and Nanobiotechnology Laboratory, IRCCS Istituto Ortopedico Rizzoli, 40136 Bologna, Italy

**Keywords:** unfolded protein response, osteosarcoma, soft tissue sarcoma, tumor microenvironment, precision medicine

## Abstract

**Simple Summary:**

Sarcomas are aggressive cancers that develop from mesenchymal tissues. They are highly diverse and often difficult to treat, with limited progress in therapy and poor outcomes for many patients, especially those with advanced or metastatic disease. To improve treatment outcomes, it is crucial to uncover details on how these tumors grow and survive. One possible weakness is the unfolded protein response (UPR), a cellular stress-response system mediated by three main proteins—IRE1, PERK, and ATF6—which help cells adapt but can also promote tumor growth, survival, and resistance to therapy. This review summarizes current knowledge about the role of the UPR in sarcomas, with particular attention to osteosarcoma. It also discusses how altering this pathway, via IRE1, PERK and ATF6 inhibition, might open new opportunities for treatment. By pulling together recent evidence, the review aims to inspire future research that could lead to more effective strategies against sarcomas.

**Abstract:**

Sarcomas are a rare and heterogeneous group of malignant tumors that pose significant clinical challenges, including delayed diagnosis, therapeutic resistance, and lack of reliable biomarkers. Despite advances in surgery and chemotherapy, effective treatment options for advanced disease remain limited, underscoring the urgent need to identify novel therapeutic vulnerabilities. The unfolded protein response (UPR), a conserved cellular stress pathway that maintains proteostasis under conditions of endoplasmic reticulum stress, has emerged as a critical modulator of cancer cell fate. By regulating protein folding, redox balance, and survival pathways, the UPR exerts a dual role in tumor biology, supporting tumor growth under stress while triggering apoptosis when stress becomes sustained or severe. In sarcomas, accumulating evidence indicates that UPR activation contributes to metabolic adaptation, angiogenesis, immune evasion, and chemoresistance. Drawing on the current literature encompassing preclinical models, recent translational research (PubMed from 2000 to 2025), and registered clinical trials, this narrative review synthesizes current knowledge on the multifaceted role of the UPR in sarcoma pathogenesis, with a particular focus on osteosarcoma. Furthermore, it explores the feasibility of UPR-targeted strategies as adjuvant or combinatorial approaches. In conclusion, this review provides an integrated and in-depth analysis of UPR-mediated mechanisms in sarcomas, offering perspectives on how targeting this pathway could accelerate the development of more effective and personalized treatments.

## 1. Introduction

Sarcomas represent a heterogeneous group of rare mesenchymal malignancies, constituting 1% of all adult solid tumors [1], with over 100 distinct histological subtypes, affecting skeletal (bone and cartilage sarcomas) or soft tissues like fat, muscles, nerves, fibrous tissues, blood vessels, or deep skin tissues (soft tissue sarcomas). Each subtype is characterized by a unique genetic and molecular landscape [2,3]. Despite advances in surgery, chemotherapy, and radiotherapy, outcomes for patients with advanced or metastatic sarcoma remain poor. This is largely due to intrinsic or acquired resistance to conventional therapies and a paucity of effective targeted treatments [3,4].

The Unfolded Protein Response (UPR), originally described as a mechanism to maintain protein folding, is now recognized as a broader regulator of cellular adaptation and stress response. It actively supports malignant progression, immune evasion, and therapy resistance [5]. While its role is well established in other tumor types [6,7,8,9,10,11], its contribution to sarcoma biology has only recently begun to emerge. Although the role of the UPR in sarcomas has yet to be fully elucidated, emerging evidence points to an association of UPR markers upregulation with poorer prognosis and increased disease aggressiveness in patients [12,13,14]. In addition, experimental findings indicate that UPR signaling contributes to key aspects of sarcoma progression, including proliferation [13,14,15,16], resistance to chemotherapy [12,17], immune evasion [13,18], and the promotion of angiogenesis [19]. Given its central role in sustaining multiple tumor-promoting processes, UPR signaling represents a promising target for therapeutic intervention in sarcomas. Pharmacologic inhibitors of the UPR are currently under investigation in other malignancies and may offer therapeutic potential for sarcoma subtypes with demonstrated UPR dependence.

This review explores current insights into UPR signaling in sarcomas, with a particular emphasis on osteosarcoma (OS), the subtype in which the role of the UPR has been most extensively studied. A detailed discussion of other stress-related pathways beyond the canonical UPR, or of endoplasmic reticulum (ER) stress responses independent of UPR activation, is beyond the scope of this work. Instead, we aim to highlight potential vulnerabilities and discuss the translational relevance of UPR targeting in these tumors. By integrating molecular data, preclinical findings, and emerging clinical perspectives, this review seeks to define the role of the UPR in sarcoma pathophysiology and its potential as a therapeutic avenue.

## 2. Unfolded Protein Response

The UPR is a highly conserved signaling pathway that includes three signaling cascades initiated by ER transmembrane proteins: IRE1 (inositol-requiring enzyme 1), PERK (double-stranded RNA-activated protein kinase (PKR)—like ER kinase) and ATF6 (activating transcription factor 6). From an evolutionary point of view, it originated with the IRE1 branch in yeast (*Saccharomyces cerevisiae*), followed by the appearance of PERK in *Caenorhabditis elegans*, and culminated in the evolution of ATF6 in mammals [20]. In mammals, this complexity is further emphasized by the presence of two ATF6 isoforms (ATF6α and ATF6β, both ubiquitously expressed, though ATF6α is far better studied) [21] and two paralogs of IRE1 (IRE1α, widely distributed, and IRE1β, primarily expressed in the respiratory and gastrointestinal tracts) [22]. Together, these variants point to a level of tissue- and cell-type-specific regulation that is still not fully understood. UPR sensors consist of three domains: a luminal domain that recognizes misfolded proteins in the ER, a transmembrane domain, and a cytosolic signal-transducing domain [23]. Together, they monitor ER homeostasis, detecting inefficacy in its protein-folding capabilities, and coordinating responses to prevent and manage the accumulation of unfolded proteins. Activation of the UPR significantly influences nearly all facets of the secretory pathway, modulating protein synthesis and entry into the ER, influencing protein folding, maturation, and quality control. Additionally, it affects protein trafficking and the clearance of misfolded proteins via the ER-associated protein degradation (ERAD) and autophagy pathways. Each UPR branch ultimately activates bZIP transcription factors, which bind DNA and foster the expression of target genes [24] (Figure 1).

ATF6α is a type II transmembrane glycoprotein with two Golgi localization signals within its ER luminal domain, where it is normally bound to the chaperone binding immunoglobulin protein (also known as glucose-regulated protein 78, GRP78, herein BiP) [25]. Upon ER stress, BiP dissociates and ATF6α is exported from the ER to the Golgi apparatus in COP-II vesicles [26], where it undergoes two sequential cleavages by S1P (site-1 protease) and S2P (site-2 protease) [27]. The resulting N-terminal fragment (ATF6α(N)) is released into the cytosol, translocates to the nucleus and binds to ER stress-response elements, stimulating the transcription of ER chaperones (e.g., BiP), XBP1 (X-box binding protein 1), enzymes involved in protein translocation, folding, maturation, and secretion, as well as enzymes responsible for degrading misfolded proteins [28].

PERK is a type I transmembrane serine/threonine kinase, which oligomerizes and trans-autophosphorylates when activated. Unlike ATF6α, PERK can be either activated by the dissociation of BiP [29,30] or by direct binding to unfolded proteins [31]. Once active, PERK phosphorylates eIF2α (α-subunit of eukaryotic initiation factor 2) [32], that on one hand, inhibits mRNAs translation to reduce protein influx into the ER, while on the other, triggers the activation of ATF4 (activating transcription factor 4). ATF4, in turn, induces the expression of genes responsible for amino acid uptake, glutathione synthesis, and oxidative stress resistance [33]. It also stimulates expression of GADD34 (growth arrest and DNA damage–inducible gene 34), which forms a complex with the catalytic subunit of PP1c (protein phosphatase 1) to dephosphorylate eIF2α, thereby establishing a negative feedback loop [34].

IRE1α is the most evolutionarily conserved branch of the UPR and has the most complex signaling pathway. It is a type I transmembrane kinase/endonuclease that, like PERK, oligomerizes and trans-autophosphorylates when activated by unfolded proteins [35] or by BiP dissociation [29,30]. Activation of its RNase activity triggers the alternative cytosolic splicing of XBP1 mRNA, removing a 26-nucleotide intron [36] and generating a transcript that encodes for XBP1s (XBP1 spliced), a transcription factor that cooperates with ATF6α(N) [37], to induce genes encoding molecular chaperones, ERAD components, as well as genes involved in ER expansion, protein import into the ER and protein folding [38]. Moreover, IRE1α can degrade selected mRNAs and miRNAs in a process known as regulated IRE1-dependent decay (RIDD), thereby lowering the protein load within the ER [39,40]. IRE1α endoribonuclease preferentially targets stem-loop structures containing a CUGCAG consensus present both in XBP1 mRNA and in RIDD target RNAs. However, recent studies reveal that RIDD can also cleave substrates lacking these motifs [41]. In all, the UPR comprises interconnected signaling pathways that dynamically modulate ER folding capacity to preserve proteostasis and support cellular function during ER stress.

When adaptive signaling is successful, PERK, ATF6α and IRE1α sustain survival by restoring ER homeostasis. However, if the stress responses persist, the UPR evolves into a “terminal UPR” that promotes apoptosis [42]. PERK, via ATF4, activates the transcription factor CHOP (C/EBP homologous protein), which suppresses anti-apoptotic BCL-2 (BCL2 apoptosis regulator) while inducing pro-apoptotic proteins like BIM (BCL-2 interacting mediator of cell death), NOXA (phorbol-12-myristate-13-acetate-induced protein 1) and PUMA (p53 upregulated modulator of apoptosis) [43]. IRE1α engages TRAF2 (tumor necrosis factor receptor-associated factor 2), activating the ASK1 (apoptosis signal-regulating kinase 1)/JNK (c-Jun N-terminal kinase) pathway and promoting BAK (BCL2 Antagonist/Killer 1)/BAX (BCL2 associated X protein)-mediated mitochondrial apoptosis. Chronic IRE1α activation also enhances RIDD-mediated mRNA decay, depleting essential proteins such as ER chaperones (e.g., BiP) and exacerbating ER stress [44]. Additionally, it induces the degradation of specific miRNAs involved in the regulation of caspase-2 expression, facilitating mitochondrial outer membrane permeabilization, while increasing pro-inflammatory proteins, such as TXNIP (thioredoxin interacting protein), which activates the NLRP3 (NLR family pyrin domain containing 3) inflammasome and cell death [45].

Beyond its role in proteostasis, UPR contributes to a wide range of biological processes [46]. Through molecular interactions, UPR proteins facilitate inter-organelle communication and signal transmission, influencing mitochondrial energy balance [47,48], cytoskeletal organization [49,50], and membrane dynamics [51]. Moreover, UPR activation plays a crucial role in the differentiation of various cell types (osteoclasts, osteoblasts, chondrocytes [52], plasma cells [53], and exocrine cells [54]), metabolic regulation [55,56], neuronal plasticity [57], and angiogenesis [58], highlighting its broader significance in cellular function and adaptation.

## 3. UPR in Cancer

Given the wide range of processes in which the UPR is involved, it is unsurprising that it is now an acknowledged hallmark of cancer [59]. Several studies have demonstrated increased expression of IRE1α, PERK and ATF6α in various tumors, including those of the lung, breast, kidney, pancreas, colon, prostate and liver. Similarly, the chaperone BiP has been found overexpressed in many tumor tissues [60]. Furthermore, somatic mutations in UPR sensor genes display distinct mutation patterns across different cancer types [61].

### 3.1. UPR and Tumor Progression

The UPR appears to play a role in every phase of tumor development. It contributes to tumor onset and progression, including neoplastic transformation, tumor growth, invasion and metastasis. Oncogenic transformation, the initial phase of tumor development, imposes a high metabolic demand on cancer cells, resulting in increased protein synthesis and translocation into the ER [62]. This process is further amplified by oncogene hyperactivation, such as MYC, RAS, or the loss of function in tumor suppressors, like p53 (tumor protein P53), PTEN (phosphatase and tensin homolog), all of which converge on UPR activation [63]. Following oncogenic transformation, UPR also promotes tumor progression by inducing epithelial–mesenchymal transition (EMT), invasion, and metastasis. PERK activation is essential for EMT, enabling cells to acquire invasive and metastatic traits through cytoskeletal remodeling and adaptation to cellular stress [64]. Similarly, IRE1α–XBP1 signaling facilitates EMT by downregulating E-cadherin and upregulating N-cadherin in breast cancer [65] and colorectal carcinoma cells [66]. In addition, IRE1α directly interacts with filamin A, an actin-binding protein, to regulate cytoskeleton remodeling and enhance cell invasiveness [49]. Furthermore, ATF6 has been shown to promote brain metastasis in breast cancer [67] while XBP1 supports invasion and proliferation by inducing MMP9 (matrix metalloproteinase-9) expression and remodelling of the extracellular matrix in human esophageal squamous cell carcinoma [68].

Beyond these cell-intrinsic effects, the UPR also influences tumor-stroma interactions, modulating the immune response, angiogenesis, and chemoresistance [69]. Depending on the intensity and duration of ER stress, as well as the tissue context, the UPR can exert both pro-tumorigenic and anti-tumorigenic effects, either promoting cancer cell survival or triggering apoptosis.

### 3.2. UPR and Lipid Metabolism

Altered lipid metabolism, including fatty acid synthesis, uptake, and oxidation, [70,71], is a well-studied feature of cancer. Recent evidence indicates that the UPR also plays a critical role in regulating metabolic and lipid homeostasis [56,72,73]. Among the UPR branches, ATF6α has been shown to promote prostate cancer progression by upregulating PLA2G4A (cytosolic phospholipase A2)-mediated arachidonic acid metabolism, leading to elevated prostaglandin production and resistance to ferroptotic cell death in tumor cells [74]. Instead, the IRE1α/XBP1s signaling axis directly supports tumor growth and survival by linking oncogenic signaling to lipid metabolic reprogramming. For example, in Myc-transformed cancer cells, XBP1s upregulates SCD1 (stearoyl-CoA desaturase 1), which catalyzes the conversion of saturated fatty acids into monounsaturated fatty acids. This process maintains membrane fluidity, supporting rapid cell proliferation, and protects cells from lipotoxic stress, thereby integrating oncogenic Myc activity with lipid metabolism. Moreover, IRE1α-dependent RIDD activity contributes to lipid metabolic reprogramming in triple-negative breast cancer (TNBC) by directly cleaving DGAT2 (diacylglycerol O-acyltransferase 2) mRNA, which encodes an enzyme critical for triacylglycerides biosynthesis and lipid droplet formation. This mechanism helps cancer cells adapt to nutrient deprivation stress [75]. Interestingly, in non-small cell lung cancer cells the IRE1α/XBP1 axis upregulates SREBP1 (sterol regulatory element-binding protein 1), which binds directly to the promoter of the MRP1 (multidrug resistance-associated protein 1) and drives its transcription. Elevated MRP1 expression facilitates drug efflux and confers resistance to cytotoxic chemotherapy, thereby establishing a link between UPR signaling, lipid metabolism, and drug resistance [76]. Consistently, pharmacological inhibition of IRE1α RNase activity with 4μ8C in hepatocellular carcinoma was found to alter lipid turnover and improve therapeutic response, sensitizing cells to doxorubicin (DOX) [77]. Additionally, BiP was found to drive chemoresistance in breast cancer by regulating fatty acids synthesis, mitochondrial transportation and oxidation [78].

### 3.3. UPR and Resistance to Therapy

Chemotherapy resistance remains one of the most significant challenges in oncology, severely limiting the long-term efficacy of anticancer treatments. This issue is exacerbated not only by the heterogeneity observed across different tumor types and among patients, but also by the substantial intratumoral heterogeneity, where distinct cellular subpopulations display variable responses to therapy. Uncovering and characterizing the molecular mechanisms underlying chemoresistance is therefore imperative for improving the prognosis and therapeutic outcomes of cancer patients. The UPR seems to play a key role in helping cancer cells survive chemotherapy, with all three UPR branches contributing to resistance [69]. For instance, in colon cancer, PERK has been shown to mediate 5-fluorouracil [79] and multidrug [80] resistance. Similarly, activation of the IRE1α–XBP1 pathway by 5- fluorouracil induces the expression of ATP-binding cassette (ABC) family drug efflux transporters such as ABCB1 (ABC subfamily B member 1), ABCC1 (alias MRP1), and ABCG2 (ABC subfamily G member 2) [81]. In ovarian cancer, inhibition of IRE1α signaling restores sensitivity to cisplatin both in vitro and in vivo [82]. Likewise, ATF6 signaling has also been implicated in promoting resistance in glioblastoma [83] and lymphoma [84], where its knockdown sensitizes tumor cells to radiotherapy or chemotherapy, respectively. Moreover, the chaperone BiP has emerged as a critical regulator of chemoresistance in pancreatic ductal adenocarcinoma [85].

### 3.4. UPR and Tumor Microenvironment

The tumor microenvironment (TME) is characterized by hostile conditions, such as acidosis, hypoxia, nutrients deficiency and oxidative stress. These stressors exacerbate protein misfolding, disrupt intracellular calcium and reactive oxygen species homeostasis, and promote persistent activation of UPR signaling [86].

Among these factors, hypoxia has a particularly profound effect on ER homeostasis, as it impairs oxygen-dependent processes such as disulfide bond formation and lipid desaturation, both necessary for proper protein folding and ER expansion [87]. To adapt, cancer cells rely heavily on the IRE1α and PERK arms of the UPR. XBP1-deficient cells display impaired survival in hypoxic conditions, indicating the need for XBP1s signaling in adaptation [88]. Mechanistically, XBP1s has been suggested to cooperate with HIF1α (hypoxia inducible-factor 1 alpha), a key mediator of hypoxic adaptation, to support cell survival under hypoxia. HIF1α is stabilized in low oxygen and activates genes involved in growth, metabolism, and angiogenesis. In TNBC, XBP1s is essential for efficient transcription of HIF1α target genes, thereby promoting tumor survival [89]. The PERK pathway is also crucial for survival under hypoxia, as PERK-deficient cells exhibit markedly reduced viability under oxygen limited conditions [90]. In addition to hypoxia, nutrient deprivation is another major stressor within the TME that perturbs ER function and sustains UPR activation. Glucose and glutamine shortage disrupt critical metabolic pathways, like the hexosamine biosynthetic pathway (HBP), which generates uridine diphosphate-N-acetylglucosamine (UDP-GlcNAc) for N-linked glycosylation (O-GlcNAcylation) and protein folding [91]. XBP1 overexpression or UPR induction enhances HBP enzyme activity, whereas XBP1 knockdown suppresses starvation-induced O-GlcNAcylation [92]. To overcome oxygen and nutrient limitation, tumors also stimulate angiogenesis, primarily through VEGF-A (vascular endothelial growth factor-A), as well as FGF (fibroblast growth factor) and PDGF (platelet-derived growth factor) [93]. In TNBC, XBP1s is required for efficient HIF1α-mediated VEGF-A expression and angiogenesis [89] and IRE1α deficiency reduces VEGF-A production and cytokines such as IL-6 (interleukin 6) and IL-8 (interleukin 8) [94]. Similarly, PERK promotes angiogenesis via ATF4, which binds the VEGF-A promoter and induces additional pro-angiogenic mediators [95]. ATF6 may also contribute, possibly through indirect regulation of XBP1 [96]. All together, these findings underscore the importance of UPR signaling in driving tumor vascularization.

While angiogenesis is a key adaptive mechanism for tumor expansion, an equally critical strategy is immune evasion. Emerging evidence suggests that UPR signaling enables cancer cells to suppress or modulate immune responses. For example, in colon adenocarcinoma and melanoma, XBP1 drives the production and secretion of cholesterol-rich small extracellular vesicles, which in turn promote the expansion and activation of myeloid-derived suppressor cells. These immunosuppressive cells inhibit T-cell activity and foster a tolerogenic TME [97]. Furthermore, ER-stressed cancer cells can secrete factors that transmit ER stress to immune cells, in a process called transmissible ER stress (TERS). Myeloid dendritic cells exposed to TERS produce tumorigenic cytokines and suppress T-cell responses. In vivo, TERS-primed dendritic cells accelerate tumor growth and reduce CD8+ T-cell infiltration. Similar effects have been observed in macrophages, where ER stress induced by cancer cells increases the expression of pro-inflammatory cytokines and stress response genes [98]. These findings highlight how UPR activation not only enables cancer cells to adapt to microenvironmental stress but also reprograms immune responses to support tumor progression (Figure 2).

## 4. UPR in Sarcomas

The UPR has been extensively characterized in various mesenchymal-derived malignancies, especially in multiple myeloma, where the IRE1α/XBP1 axis is central to plasma cell survival under proteotoxic stress [34]. Over the past decade, there has been growing interest in exploring UPR mechanisms in sarcomas, also of mesenchymal origin, raising the possibility that molecular insights from other mesenchymal tumors may be translationally relevant to understanding and targeting sarcoma biology.

Sarcomas are highly aggressive tumors characterized by rapid growth, local invasiveness, and an early tendency for metastatic dissemination. A hallmark of sarcomas is their high cellular plasticity, shaped by genetic and epigenetic alterations, adaptation to the TME, and therapeutic selection pressures. Together, these factors drive tumor progression and relapse. Importantly, sarcomas represent a miscellaneous group of neoplasms with distinct genetic abnormalities, depending on the subtype and patient. Some, such as Ewing’s sarcoma (ES) [99], rhabdomyosarcoma (RMS) [100] and liposarcoma [101], harbor chromosomal translocations and fusion genes, while others, such as OS [102], liposarcoma [103] and leiomyosarcoma [104], display genome complexity and aneuploidy. Clinically, sarcomas are further complicated by their poor response to chemotherapy and radiotherapy, and frequent multidrug resistance. The absence of highly specific biomarkers continues to limit early detection and the development of effective targeted therapies, underscoring the urgent need for improved strategies in their management.

Interestingly, UPR activation can be triggered by genomic alterations, including aneuploidy [105] and fusion oncogenes [14], both of which disrupt protein homeostasis and induce cellular stress. Given that many sarcomas display high levels of genomic instability, chromosomal aneuploidy, and recurrent gene fusions, it is plausible that UPR activation represent a biologically relevant process in these tumors. This may occur either as a direct consequence of defined genetic abnormalities or as a broader adaptive mechanism in subtypes lacking clear molecular drivers, further emphasizing the potential role of the UPR in sarcomas biology.

### 4.1. UPR in Bone Sarcomas

#### 4.1.1. UPR in Osteosarcoma

OS is the most prevalent bone cancer and is characterized by an aggressive clinical course, with a strong tendency to metastasize, especially to the lungs. Despite ongoing research and therapeutic advancements, patient prognosis has remained largely unchanged over the past four decades [106]. Many features of the TME discussed previously are also highly relevant to OS, supporting the hypothesis that the UPR may play a critical role in sustaining OS aggressiveness and therapeutic resistance.

Recent studies have shown that UPR markers are upregulated in OS and may serve as prognostic indicators, as they are associated with more aggressive disease phenotypes (Table 1). In particular, Shi et al. reported enhanced expression of BiP, XBP1s, ATF4 and ATF6α in OS samples compared to normal tissues [13]. Clustering of patients revealed that increased UPR activity associates with high tumor cell proliferation, low immune cell infiltration, reduced immunotherapy response, and poor survival [13]. Similarly, Zhang et al. demonstrated that expression patterns of UPR-related genes could predict tumor-associated immune infiltration and prognosis of OS patients [107]. At the single-cell level, transcriptomic analysis revealed that ATF6α and its downstream targets are enriched in OS cells and correlate with WNT and TGF-β (transforming growth factor beta) pathways, both known to promote OS progression [108,109], and with poor prognosis [15]. Functionally, ATF6 has emerged as a critical mediator of OS aggressiveness. Its inhibition, either by genetic silencing or pharmacological repression (with the small molecule ceapin-7) reduced HOS cell viability, proliferation and colony formation [15]. Proteomic analysis further revealed that ATF6α-dependent upregulation of ER chaperones, like BiP, HSP90B1 (heat shock protein 90 beta family member 1) and calreticulin in OS tissues, compared to the soft tissue callus used as a non-cancerous control. Among these, BiP was associated to resistance to chemotherapeutic agents like DOX and platinum-based drugs [12]. Additionally, BiP loss sensitized OS cells to the proteasome inhibitor bortezomib (BTZ) by upregulating ATF4 and CHOP [110]. In a retrospective study on OS patients treated with cisplatin, DOX, and methotrexate, high ATF6α(N) expression correlated with PDI (Protein disulfide isomerase) and BiP, metastatic disease, and poor response to chemotherapy [17]. Supporting these findings, in vitro experiments demonstrated that inhibition of ATF6α, but not IRE1α and PERK, sensitized OS cells to chemotherapy-induced death, highlighting its potential role in therapeutic resistance [17].

Beyond ATF6, the IRE1 branch of the UPR also contributes to OS progression. XBP1s is overexpressed in OS cells compared to osteoblasts [12] and has been linked to disease progression and low tumor necrosis rate [111]. Knockdown of XBP1 in MG63 and U2OS cells delayed cell cycle progression and decreased viability via PIK3R3 (phosphoinositide-3-kinase regulatory subunit 3)/mTOR (mechanistic target of rapamycin kinase) pathway, particularly under hypoxic conditions [111]. Moreover, XBP1 promoted tumor growth and metastasis by binding to the GNL3 (G protein nucleolar 3—a nucleolar protein involved in cell proliferation, cell cycle, and invasion) promoter and fostering its expression [112].

PERK signaling also plays a dual role in OS, contributing to both tumor progression and modulation of the immune infiltrate within the TME. PERK inhibition reduces tumor growth and immunosuppression [18], while one of it downstream targets, STC2 (stanniocalcin 2), is upregulated in OS tissues and correlates with poorer survival outcomes [113].

**Table 1 cancers-17-03489-t001:** Clinical correlation between UPR markers and OS progression.

UPR Markers	Datasets	Normal Tissue Control	Clinical Correlation	Total Number of Patients/Samples	Reference
BiP, XBP1s, ATF4 and ATF6α	GSE99671, GSE126209, GSE21257, TARGET, Zhengzhou, TCGA sarcoma dataset	Yes	Tumor growth, low immune infiltration, reduced immunotherapy response, poor survival	512	[13]
UPR related genes (STC2, PREB, TSPYL2, and ATP6V0D1)	GSE21257, TARGET	Yes	immune infiltration and prognosis	138	[107]
ATF6α	GSE152048 and GSE162454	Yes	OS progression and poor prognosis	110,000 single cells from 17 samples	[15]
BiP	Maharaj Nakorn Chiang Mai Hospital	Yes	Chemoresistance	20	[12]
ATF6α(N)	Phoenix Children’s Hospital	Yes	Metastatic disease and poor response to chemotherapy	40	[17]
XBP1s	Shanghai Jiao Tong University Affiliated Sixth People’s Hospital	Yes	Disease progression and low tumor necrosis rate	40	[111]
STC2	GSE21257, GSE33382 and TARGET	Yes	Poor survival	225	[113]

In addition to its direct effects, the UPR operates within a dynamic regulatory network, interacting with multiple intracellular signaling pathways. For example, ER stress induces the long non-coding RNA LINC00629, which promotes OS progression via the KLF4 (Kruppel-like factor 4)-LAMA4 (laminin subunit alpha 4) axis [114]. Similarly, upregulation of GADD45GIP1 (growth arrest and DNA damage-inducible gamma interacting protein 1) activates the PERK-eIF2α axis, promoting ER stress and OS progression [115]. Furthermore, SREBP1 overexpression enhances PERK-mediated ER stress and apoptosis, suggesting a synergistic interaction between metabolic and stress-response pathways in OS [116]. Interestingly, in U2OS cells, the UPR directly impacts circadian rhythms and cell survival. Activation of the UPR, primarily through the PERK- inducible miR-211 axis, causes a phase shift in circadian oscillations and suppresses core regulators like BMAL1 (basic helix-loop-helix ARNT like 1) and CLOCK (Clock Circadian Regulator). This suppression of BMAL1 is essential for UPR-dependent inhibition of protein synthesis, enabling OS cells to adapt to ER stress [117].

Reflecting this complex interplay, transcriptomic analyses have identified a strong increase in UPR-related genes in high-risk OS groups. For instance, a risk score model based on sixteen glutamine metabolism-related genes, revealed significant UPR enrichment in the high-risk subgroup [118]. Similarly, another investigation employing m6A-related lncRNA signatures revealed a strong association with increased UPR activity in high-risk patients [119].

Altogether, these findings underscore a pro-tumorigenic role of the UPR in OS, promoting cellular proliferation, invasion, metastasis, immune evasion, and resistance to therapy (Figure 3).

#### 4.1.2. UPR in Ewing’s Sarcoma

ES is a prevalent pediatric bone tumor characterized by the EWS (Ewing sarcoma breakpoint region 1)/FLI1 (friend leukemia integration 1 transcription factor) gene fusion. In contrast to OS, the role of the UPR in ES remains poorly characterized. To date, research in this area is extremely limited, with only a single study directly investigating how UPR signaling influences ES biology. Tanabe et al. identified over 2000 proteins regulated by this fusion gene, among which the XBP1 pathway was highlighted as critical for tumor viability. High mRNA expression of XBP1 was confirmed in ES cell lines and patient surgical samples, and silencing XBP1 significantly suppressed ES cell proliferation and viability [14]. Among the IRE1-XBP1 inhibitors tested, toyocamycin was the most effective. In vivo experiments showed that toyocamycin significantly reduced tumor volume and weight in mouse xenograft models and increased cellular apoptosis [14]. Future studies should explore the mechanisms behind XBP1 activation. Special focus should be given to the regulatory role of the EWS/FLI1 fusion gene, which is central to this cancer, since current data reveal a major gap in our understanding. Additionally, more research is needed to elucidate whether the UPR exerts a pro-tumorigenic, anti-tumorigenic, or context-dependent role in ES, as well as to clarify its potential therapeutic relevance.

### 4.2. UPR in Soft-Tissue Sarcomas

Soft tissue sarcomas originate in the soft tissues of the body and can develop in virtually any anatomical site. Their rarity and biological heterogeneity complicate early detection and treatment, emphasizing the urgent need to identify new therapeutic targets [120].

RMS, the most common soft tissue pediatric cancer, exhibits basal activation of the IRE1 and PERK branches of the UPR. Pharmacological inhibition of these pathways with MKC8866 (an IRE1α inhibitor, used at 10 μM, 20 μM and 40 μM) and AMGEN44 (a PERK inhibitor, used at 2 μM, 5 μM and 10 μM) resulted in a marked reduction in RMS cell viability, proliferation, and colony-forming ability, with the two highest concentrations eliciting the most pronounced effects. Notably, combined inhibition (MKC8866 used at 20 μM and AMGEN44 used at 2 μM) produced stronger anti-proliferative effects than either agent alone, suggesting a potential synergistic impact on RMS cells. Subtype-specific differences in sensitivity were observed. Alveolar RMS cells exhibited higher responsiveness to IRE1α inhibition, whereas embryonal RMS cells were more susceptible to PERK inhibition. Mechanistically, UPR blockade induced cellular senescence, contributing to decreased proliferation. Importantly, both inhibitors showed minimal toxicity in non-malignant cells, underscoring their potential as selective therapeutic agents for RMS [16]. Further confirmation of these results in physiologically relevant 3D and in vivo models will be essential to establish their translational significance.

Another study reported that the expression of key UPR components (BiP, XBP1s, and IRE1α) was significantly associated with RMS. Specifically, BiP expression correlated with lymph node involvement, while both BiP and XBP1s were linked to all four RMS subtypes, including alveolar, embryonal, pleomorphic, and sclerosing/spindle cell RMS. In contrast, IRE1α expression was associated with alveolar, pleomorphic, and embryonal RMS, with the highest levels observed in alveolar subtype. Furthermore, XBP1s expression was correlated positively with distant metastasis in alveolar RMS and with tumor size in pleomorphic RMS. These findings suggest that activation of the IRE1α/XBP1s axis may contribute to disease aggressiveness and subtype-specific progression [121].

Overall, current evidence points toward a predominantly pro-tumorigenic role for the UPR in RMS, with the IRE1α/XBP1s and PERK branches contributing to proliferation, metastasis, and subtype-specific tumor progression. Nonetheless, these findings remain preliminary, and further investigations are needed to comprehensively define the functional and molecular mechanisms through which the UPR influences RMS pathophysiology. Such investigations will be crucial not only for clarifying the functional role of the UPR in RMS but also for assessing the therapeutic potential of UPR-targeted strategies in this malignancy.

Whereas the UPR appears to sustain proliferation and subtype-specific progression in RMS, its role in angiosarcoma extends to the regulation of angiogenesis, accentuating the context-dependent nature of UPR signaling across different sarcomas. Angiosarcomas exhibit a distinctive UPR signature, characterized by elevated BiP and PERK expression alongside reduced IRE1α protein levels. BiP is essential for maintaining VEGFR2 (VEGF receptor 2) stability and endothelial cell functions; its depletion severely impairs angiogenesis and can even cause regression of vascular structures. Elevated PERK expression, in turn, promotes the accumulation of unspliced XBP1, which has been linked to increased proliferation and malignancy. Although IRE1α protein levels are attenuated, its kinase activity remains functionally relevant, as it governs the splicing of XBP1, a central regulator of angiogenic signaling. Consistently, pharmacological inhibition of IRE1α kinase suppresses in vitro angiogenesis [19].

Beyond angiosarcoma, limited evidence suggests that UPR signaling may also play a role is Kaposi’s sarcoma, primarily through interactions with HHV-8 (human herpesvirus 8), the causative agent of this cancer. Preliminary studies have explored the involvement of XBP1 signaling in this context [122], suggesting a potential, though yet unexplored, link between the UPR activation and Kaposi’s sarcoma pathogenesis. Moreover, malignant peripheral nerve sheath tumors (MPNSTs) display elevated UPR markers, including BiP, eIF2α and XBP1s, compared with normal nerve tissue [123].

Overall, while current understanding of the role of the UPR in soft tissue sarcomas remains incomplete, emerging data suggest that it may represent a promising therapeutic target. Given the context-specific nature of UPR signaling, future research should aim to dissect its functional roles within individual sarcoma subtypes to better define its contribution to tumor progression and treatment response.

In other sarcoma subtypes, like leiomyosarcoma, liposarcoma, synovial sarcoma and chondrosarcoma, the role of the UPR remains unexplored. Notably, in chondrosarcoma, only a single study reports that PRP-1 (proline-rich polypeptide), a toll-like receptor ligand, increases the expression of PERK, eIF2α, ATF4, CHOP, ATF6, IRE1α, and XBP1 [124]; however, no studies have yet investigated the functional role of the UPR in this tumor.

## 5. Targeting the UPR in Cancer

The evidence discussed thus far underscores a predominantly pro-tumorigenic role of the UPR in sarcomas. Nevertheless, it is essential to acknowledge the dual nature of the UPR: under conditions of sustained ER stress, its activation can shift toward the “terminal UPR,” which drives apoptosis in malignant cells.

Several agents, such as MO-OH-Nap, an α-substituted tropolone, induce UPR-mediated apoptotic death in several OS cell lines [125], while psoralen, a natural active component of *Psoralea corylifolia*, triggers ATF6α/CHOP-dependent apoptosis [126]. Sdox [127], a DOX analog, and clemastine [128], an antagonist of histamine H1 receptor, also activate ER stress-dependent apoptotic pathways in OS cells. Furthermore, compounds like CB-5083 activate both PERK and IRE1α to induce apoptosis in OS cells [129], and traditional medicines such as tongguanteng [130], or active peptides like mellitin [131] activate IRE1α/XBP1/CHOP axis to promote OS cell death. EUR, an active compound in *Eurycoma longifolia* Jack root, inhibits OS growth by destabilizing BiP mRNA and blocking its transcription [132]. Moreover, propofol, a common anesthetic, suppresses OS proliferation and invasion through ER stress induction and synergizes with DOX [133]. Similarly, HMnO_2_@CaO_2_, a manganese dioxide nanoplatform with incorporated calcium peroxide, induces both ER stress and ferroptosis, showing promising antitumor activity [134]. Knockout of the DNMT2/TRDMT1 (tRNA aspartic acid methyltransferase 1) gene sensitizes cells to ER stress-induced apoptosis by impairing UPR activation upon DOX treatment [135]. In addition, β-elemonic acid, an active component extracted from *Boswellia carterii* bird, activates the PERK/eIF2α/ATF4 branch, stimulating CHOP-regulated apoptosis and Ca^2+^-mediated caspase activation [136], while inhibition of GGDPS (geranylgeranyl diphosphate synthase) with RAM2061 induces UPR markers and apoptosis in both OS and ES cells [137].

Similar vulnerabilities are observed in other sarcomas: the IRE1α inhibitor MKC3946 sensitizes leiomyosarcoma cells to apoptosis [138], and proteasome inhibition with carfilzomib enhances nutlin-3-induced apoptosis in liposarcoma by activating the ATF4/CHOP/NOXA axis [139]. Moreover, nelfinavir, a clinically approved HIV (human immunodeficiency virus) protease inhibitor, blocks S2P activity, leading to inhibition of SREBP1 and ATF6 signaling and subsequent apoptosis [140]. In MPNST, BTZ-driven UPR activation enhances the synergistic antitumor efficacy observed when combined with oHSV (oncolytic herpes simplex virus-1) [141], while ER-stress-inducing agents such as thapsigargin, tunicamycin or the HSP90 inhibitor IPI-504 cause UPR overexpression leading to cell death [123]. In RMS, both the HSP70 (heat shock 70 kDa protein) inhibitor MAL3-101 and the p97 (valosin containing protein) ATPase inhibitor CB-5083 trigger robust UPR activation, marked by increased eIF2α phosphorylation, ATF4 and CHOP upregulation, and XBP1 splicing, thereby promoting tumor cell death [142]. Recently, nanostructures and nanoformulated drugs have emerged as promising tools to modulate or intercept the UPR in cancer cells. In glioblastoma, the PDI inhibitor CCF642, especially when delivered via albumin-based nanoparticles, was shown to activate apoptosis-inducing UPR, downregulate PERK signaling, and restore sensitivity to temozolomide [143]. In lung-carcinoma cells, the UPR is triggered by denatured proteins in the nanoparticle corona, which recruits the chaperone Hsp90ab1 to the particle surface and up-regulates downstream pathways that promote EMT. Blocking Hsp90ab1 with geldanamycin prevents UPR-induced activation and EMT, pointing to the UPR-Hsp90ab1 axis as a driver of tumor cell reprogramming [144].

In addition, it is noteworthy that preclinical studies in cancers other than sarcomas have provided compelling evidence that pharmacological inhibition of the UPR can exert significant antitumor effects, particularly when combined with standard therapies. Inhibiting the IRE1α/XBP1 axis with Z4P enhances temozolomide efficacy in glioblastoma [145], while 4µ8C and STF-083010 potentiate 5-fluorouracil in colorectal cancer [146] and show strong anti-myeloma activity [147]. Local MKC8866 delivery in glioblastoma prolongs survival when combined with radiochemotherapy [148]. Targeting PERK has yielded similarly promising results: GSK2606414 potentiates oridonin in small-cell lung cancer [149], enhances sorafenib activity in hepatocellular carcinoma [150], and, notably, boosts PD-1 immunotherapy in murine sarcoma [151]. GSK2656157 synergizes with 5-fluorouracil in colon cancer [79], reduces hepatocellular carcinoma growth [152], and, in melanoma, improves responses to mRNA vaccination [153]. More recently, PERK inhibition with HC-5404 enhanced VEGFR-targeted therapy in renal carcinoma, achieving durable tumor control [154]. Collectively, these findings underscore the translational relevance of UPR inhibitors as partners to established therapies, immunotherapies, and resistance-modulating approaches.

Several clinical trials (https://clinicaltrials.gov/) are currently investigating therapeutic strategies targeting different branches of the UPR and related stress pathways across a variety of malignancies, even though clinical trials in sarcomas remain scarce (Table 2). Compounds such as BOLD-100 and NKP-1339 [155], BiP inhibitors, are being tested both as single agents and in combination with standard chemotherapeutic regimens (e.g., FOLFOX, DOX) in gastrointestinal cancers, soft tissue sarcomas, and other solid tumors. Inhibition of PERK is being explored with agents such as HC 5404-FU and NMS-03597812, the latter evaluated in combination with dexamethasone for multiple myeloma and as monotherapy for acute myeloid leukemia. Targeting the IRE1α RNase domain by MKC8866 is currently in trials in breast cancer and solid tumors, either in combination with Abraxane or with standard-of-care regimens [156]. In parallel, proteasome inhibition with IXAZOMIB is under investigation in non-hematologic malignancies, including soft tissue sarcomas.

Collectively, these studies reflect the growing clinical interest in modulating ER stress and UPR-related pathways as potential therapeutic avenues in both hematologic and solid tumors.

## 6. Conclusions and Future Perspectives

The growing evidence for therapeutic targeting of UPR across various tumor types supports the extension of these strategies to sarcomas. In OS, where scientific data are more robust, investigating UPR modulation could address critical clinical challenges including relapse, therapy resistance, and metastatic dissemination, and integrating UPR-targeted approaches with standard-of-care treatments, like DOX and cisplatin, may enhance therapeutic efficacy and improve patient outcomes. The UPR dual capacity to promote both cell survival and apoptosis necessitates precise interventions that shift this balance toward tumor suppression. Inhibiting adaptive UPR branches may limit tumor growth and resistance, while exacerbating ER stress to trigger terminal UPR activation offers a complementary approach to induce tumor apoptosis. The association between elevated UPR marker expression and poor prognosis in OS patients further highlights the potential for precision medicine approaches, where stratifying patients based on UPR activity could identify those most likely to benefit targeted therapies, while minimizing unnecessary toxicity. However, translating these insights into clinically effective interventions requires further elucidation of the underlying molecular mechanisms. Moreover, comprehensive investigations are required to elucidate the functional relevance of the UPR within specific sarcoma subtypes. Future studies should begin by systematically characterizing standardized panels of UPR biomarkers, including IRE1, ATF6, PERK, BiP, and XBP1, to define their expression patterns and activation states across different sarcoma models. Subsequent efforts should aim to clarify how these signaling components contribute to distinct aspects of tumor biology, such as proliferation, survival, invasion, and therapeutic resistance, and to uncover the molecular pathways mediating these effects. Ultimately, in vivo studies will be essential to validate these findings and to assess the therapeutic potential of combinational strategies that target the UPR alongside conventional or novel anticancer treatments.

## Figures and Tables

**Figure 1 cancers-17-03489-f001:**
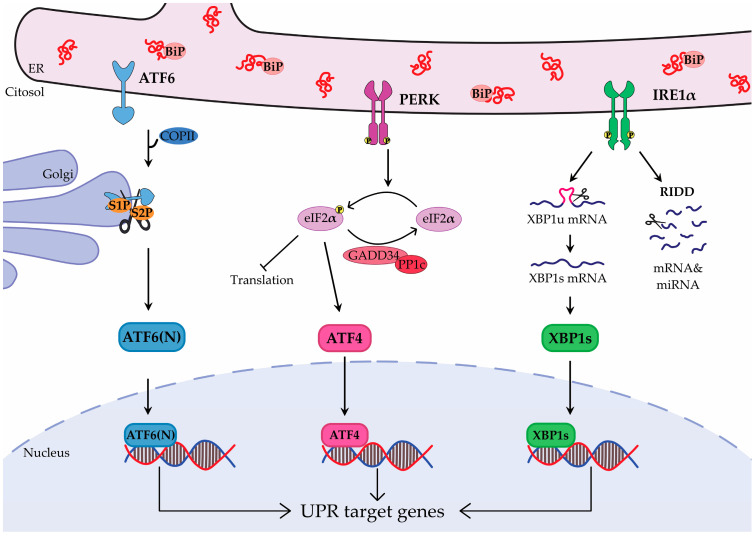
Schematic representation of the Unfolded Protein Response (UPR) signaling. Unfolded or misfolded proteins in the endoplasmic reticulum (ER) activate the three branches of the UPR: ATF6 (activating transcription factor 6), PERK (double-stranded RNA-activated protein kinase (PKR)–like ER kinase) and IRE1 (inositol-requiring enzyme 1). ATF6 is activated when BiP (binding immunoglobulin protein) dissociates from its ER luminal domain and is transported to the Golgi for proteolytic cleavage by S1P (site-1 protease) and S2P (site-2 protease), which generates ATF6(N) (ATF6 N-terminal fragment). PERK is activated either by BiP dissociation or by direct binding of unfolded proteins. It phosphorylates eIF2α (α-subunit of eukaryotic initiation factor 2), which reduces protein translation and activates ATF4 (activating transcription factor 4). GADD34 (growth arrest and DNA damage–inducible gene 34) and PP1c (protein phosphatase 1) form a complex to dephosphorylate eIF2α. IRE1, activated by unfolded proteins or by BiP dissociation, cleaves XBP1 (X-box binding protein 1) mRNA producing XBP1s (XBP1 spliced). It also mediates the selective breakdown of mRNAs and miRNAs via regulated IRE1-dependent decay (RIDD). ATF6(N), ATF4 and XBP1s translocate to the nucleus, where they trigger the expression of target genes.

**Figure 2 cancers-17-03489-f002:**
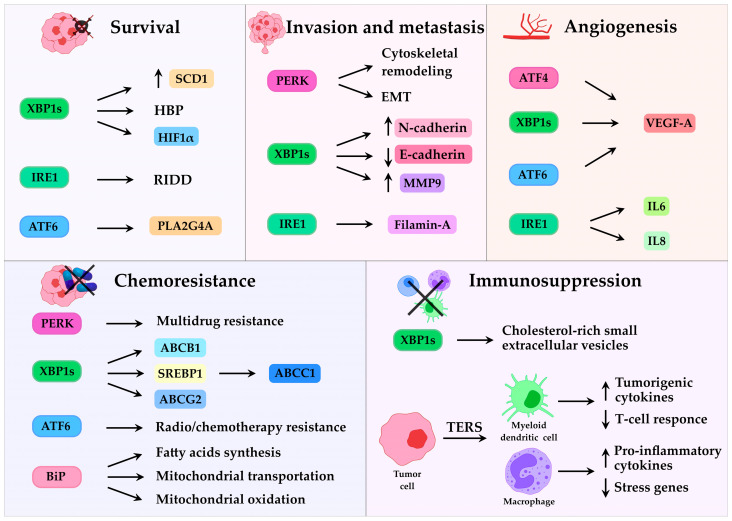
Diagram of Unfolded Protein Response (UPR)-mediated signaling in tumor cells, highlighting how IRE1 (inositol-requiring enzyme 1), PERK (double-stranded RNA-activated protein kinase (PKR)–like ER kinase), and ATF6 (activating transcription factor 6) pathways facilitate cancer survival, progression and resistance to treatment. XBP1s (X-box binding protein 1 spliced) upregulates SCD1 (stearoyl-CoA desaturase 1), to prevent lipotoxic stress, stimulates the hexosamine biosynthetic pathway (HBP) to support survival under starvation, and cooperates with HIF1α (hypoxia inducible-factor 1 alpha) to sustain cell survival under hypoxia. Regulated IRE1-dependent decay (RIDD) aids adaptation to nutrient deprivation, while ATF6 upregulates PLA2G4A (cytosolic phospholipase A2)-mediated arachidonic acid metabolism, to promote resistance to ferroptotic cell death. During invasion and metastasis, PERK drives epithelial–mesenchymal transition (EMT) and metastasis through cytoskeletal remodeling, while XBP1s induces EMT by downregulating E-cadherin and upregulating N-cadherin and upregulates MMP9 to support extracellular matrix remodeling and invasion. Moreover, IRE1 directly interacts with filamin A to enhance invasiveness. To promote angiogenesis, ATF4 (activating transcription factor 4), XBP1s and ATF6 upregulate VEGF-A, while IRE1 augments IL6 (interleukin 6) and IL8 (interleukin 8) expression. Chemoresistance involves all three branches: PERK promotes multidrug resistance, while XBP1s enhances drug efflux via ABCB1 (ABC subfamily B member 1), ABCC1 (alias, MRP1, multidrug resistance-associated protein 1—through SREBP1 (sterol regulatory element-binding protein 1)) and ABCG2 (ABC subfamily G member 2). AFT6 contributes to radio- and chemoresistance, while BiP (binding immunoglobulin protein) drives chemoresistance by regulating fatty acid synthesis, mitochondrial transportation and oxidation. Finally, UPR activation fosters immune evasion, as XBP1s drives a tolerogenic tumor microenvironment via cholesterol-rich small extracellular vesicles, while transmissible ER stress (TERS) reprograms myeloid dendritic cells and macrophages to produce tumorigenic and pro-inflammatory cytokines and suppress T-cell responses.

**Figure 3 cancers-17-03489-f003:**
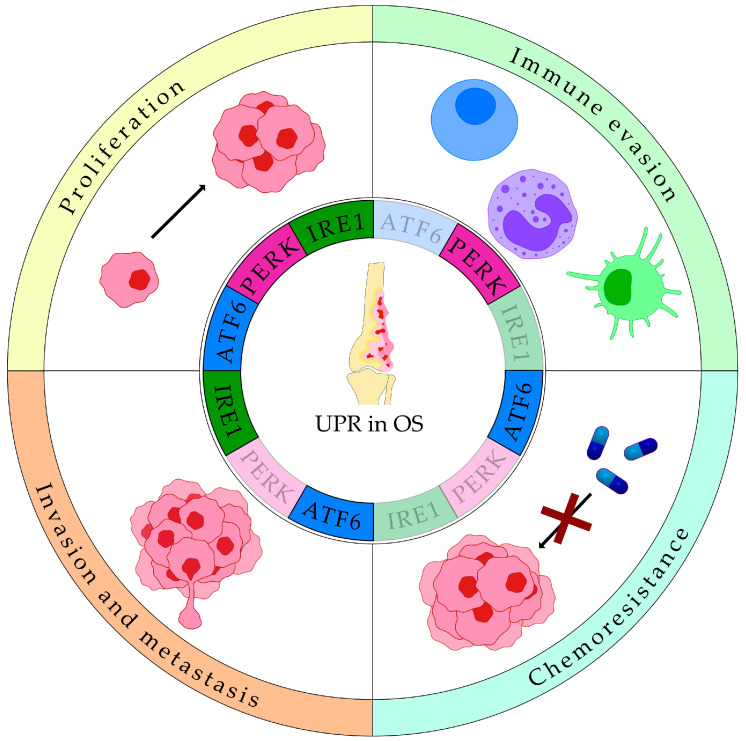
Roles of the Unfolded Protein Response (UPR) in osteosarcoma (OS) survival and progression. All three UPR branches, ATF6 (activating transcription factor 6), PERK (double-stranded RNA-activated protein kinase (PKR)–like ER kinase), and IRE1 (inositol-requiring enzyme 1), contribute to the regulation of tumor proliferation. Beyond this shared role, PERK also mediates immune evasion, IRE1 and ATF6 promote invasion and metastasis, and ATF6 further drives chemoresistance.

**Table 2 cancers-17-03489-t002:** Drug targeting the UPR currently in clinical trials.

Drug	Combination Drug Therapy	Target	Tumor	Clinical Trial ID	Clinical Trial Phase
BOLD-100	FOLFOX Chemotherapy	BiP	Gastric, Pancreatic, Colorectal Cancer and Cholangiocarcinoma	NCT04421820	Phase 1 & 2
BOLD-100	DOX	BiP	Soft tissue sarcomas	NCT07027423	Phase 1
NKP-1339	None	BiP	Solid tumors	NCT01415297	Phase 1
HC 5404-FU	None	PERK	Renal cell carcinoma, gastric cancer, metastatic breast cancer, small cell lung cancer, and other solid tumors (e.g., non-small cell lung cancer, colorectal cancer, carcinoma of unknown primary)	NCT04834778	Phase 1
NMS-03597812	Dexamethasone	PERK	Multiple Myeloma	NCT05027594	Phase 1
NMS-03597812	None	PERK	Acute Myeloid Leukemia	NCT06549790	Phase 1
ORIN 1001 (MKC8866)	Abraxane	IRE1 RNase	Breast cancer and other solid tumors	NCT03950570	Phase 1 & 2
ORIN 1001 (MKC8866)	Standard of Care	IRE1 RNase	Solid tumors	NCT05154201	Phase 1 & 2
IXAZOMIB	None	Proteasome	Nonhematologic Malignancies including Soft Tissue Sarcomas	NCT00830869	Phase 1

## Data Availability

No new data were created or analyzed in this study. Data sharing is not applicable to this article.

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
