# Peer review of "The Unfolded Protein Response in Sarcomas: From Proteostasis to Therapy Resistance"

_cancers, 2025, doi:10.3390/cancers17213489_

Round 1

Reviewer 1 Report

Comments and Suggestions for Authors

The present manuscript reviews the current knowledge of the unfolded protein response (UPR) pathway and its role in sarcoma development, with a particular focus on osteosarcoma. The manuscript is well-structured, starting with a description of the UPR molecular mechanisms, followed by its implications in general tumor development, the potential role in sarcoma biology and finishing with the consideration of UPR as an appealing target for anticancer therapies that promote UPR to its “terminal UPR” mode and potential induction of tumor cell apoptosis, or for the development of drug that inhibit UPR pathways.

The text is clear and easy to follow, supported by clear figures and tables. References are in correspondence with the topic.

I recommend this manuscript for acceptation without any major corrections. Potentially, the quality of figures could be improved.

Author Response

Comments 1: The present manuscript reviews the current knowledge of the unfolded protein response (UPR) pathway and its role in sarcoma development, with a particular focus on osteosarcoma. The manuscript is well-structured, starting with a description of the UPR molecular mechanisms, followed by its implications in general tumor development, the potential role in sarcoma biology and finishing with the consideration of UPR as an appealing target for anticancer therapies that promote UPR to its “terminal UPR” mode and potential induction of tumor cell apoptosis, or for the development of drug that inhibit UPR pathways.

Response 1: We are grateful to the reviewer for their positive feedback.

Comments 2: I recommend this manuscript for acceptation without any major corrections. Potentially, the quality of figures could be improved.

Response 2: As suggested by the reviewer, we have added another Figure (now figure 2 – page 8) and have improved the quality of Figure 2 (now figure 3 – page 12).

Reviewer 2 Report

Comments and Suggestions for Authors

This narrative review tackles a clinically important and under-synthesized topic: how the UPR shapes sarcoma biology and therapy resistance, with emphasis on OS. The manuscript is timely and generally well organized, but it requires substantial revisions:

Title, authors, affiliations, notes:

-          The author line contains formatting artifacts (duplicate “and” and misplaced equal-contribution markers): “§and Margherita Cortini 1,*§”. Please correct author separators and symbol placement, and standardize the equal-contribution footnote.

Simple Summary:

- Consider (i) defining UPR very briefly in lay terms once, (ii) adding one concrete example of a therapeutic angle (e.g., PERK/IRE1 inhibitors) to anchor relevance, and (iii) trimming repeated phrases about “opening new avenues.”

Abstract

-          Grammar: “…pave the way for to complement existing treatments to personalized medicine approaches.” Suggest: “pave the way for personalized-medicine approaches that complement existing treatments.”

-          Please state scope and method (e.g., narrative review; databases/timeframe; non-systematic selection) so readers can gauge coverage and bias.

Introduction:

-          Consider adding one sentence with epidemiology/incidence to frame rarity and burden.

-          End the section with an explicit aims paragraph (what the review covers, why the emphasis on OS, what is outside scope).

UPR biology (Figure 1):

-          The molecular overview is helpful. Please fix wording: “signal-transducting domain” → “signal-transducing domain.”

-          Ensure the figure legend defines all acronyms introduced here (ATF6(N), ERAD, etc.) and cross-reference to the Abbreviations section.

-          Minor style: consistently format “bZIP” (currently appears as “b-ZIP”).

UPR in cancer (angiogenesis, immune evasion, TME):

-          Consider adding a schematic that links UPR branches → downstream effectors → phenotypes (angiogenesis, MDSCs, T-cell dysfunction) to aid non-specialist readers.

UPR in sarcomas:

4Bone sarcomas — Osteosarcoma:

- Typos/clarity: “transforming growth factor beta (TFG-)” should be TGF-β; please correct throughout.

-          I suggest adding a summary table for OS (marker, model/patient cohort, readout, clinical correlation, n).

Ewing’s sarcoma:

-          Consider briefly noting whether EWS/FLI1 mechanistically engages UPR sensors directly/indirectly (as a hypothesis for future work).

Soft-tissue sarcomas (RMS, angiosarcoma, Kaposi’s)

-          RMS: Suggest adding exposure/dose context and whether effects persisted in 3D or in vivo models if available.

- Consider clarifying whether reduced IRE1α protein but retained kinase activity implies selective axis engagement in endothelium.

- Add terse notes on leiomyosarcoma, liposarcoma, synovial sarcoma, MPNST, chondrosarcoma (even to state negative/limited evidence).

Targeting the UPR in cancer (and Table 1)

-          The preclinical compendium is useful and shows dual strategies (inhibit adaptive UPR vs push terminal UPR).

Critical table issues to fix:

-          ORIN1001 vs MKC8866: The table labels “ORIN 1001 (MKC8866),” suggesting they are the same drug; they are distinct IRE1 RNase inhibitors and should not be conflated. Please correct drug names/rows and citations.

Conclusions and Future Perspectives:

- Consider a closing research roadmap: (i) harmonized biomarker panels (ATF6α(N)/BiP/XBP1s), (ii) subtype-specific functional studies beyond OS, (iii) combination hypotheses (UPR modulators + DOX/BTZ/IO) with patient stratification by UPR activity.

Comments on the Quality of English Language

English must go through an extensive review process.

Author Response

Comments 1: This narrative review tackles a clinically important and under-synthesized topic: how the UPR shapes sarcoma biology and therapy resistance, with emphasis on OS. The manuscript is timely and generally well organized, but it requires substantial revisions: 

Response 1: We thank the reviewer for the general constructive comments. We hope that the quality of the review has now been improved by the revision.

Comments 2: Title, authors, affiliations, notes:

-         The author line contains formatting artifacts (duplicate “and” and misplaced equal-contribution markers): “§and Margherita Cortini 1,*§”. Please correct author separators and symbol placement, and standardize the equal-contribution footnote.

Response 2: We have corrected the artifacts and typos.

Comments 3: Simple Summary:

-    Consider (i) defining UPR very briefly in lay terms once, (ii) adding one concrete example of a therapeutic angle (e.g., PERK/IRE1 inhibitors) to anchor relevance, and (iii) trimming repeated phrases about “opening new avenues.”

Response 3: As suggested by the reviewer, we added a brief definition of the UPR “a cellular stress-response system mediated by three main proteins – IRE1, PERK, ATF6 – which help cells adapt”, an example of therapeutic approach “via IRE1, PERK and ATF6 inhibition” and cutting the phrase “highlight new directions for targeted therapies” (page 1).

Comments 4: Abstract

-         Grammar: “…pave the way for to complement existing treatments to personalized medicine approaches.” Suggest: “pave the way for personalized-medicine approaches that complement existing treatments.”

Response 4: We have changed the sentence accordingly, as suggested by the reviewer.

Comments 5: -          Please state scope and method (e.g., narrative review; databases/timeframe; non-systematic selection) so readers can gauge coverage and bias.

Response 5: As suggested by the reviewer, we have expanded the abstract with the methods, the used databases and a better description of the scope of the manuscript. The text now states:

Drawing on the current literature encompassing preclinical models, recent translational research (PubMed from 2000 to 2025), and registered clinical trials (clinicaltrials.gov), this narrative review synthesizes current knowledge on the multifaceted role of the UPR in sarcoma pathogenesis, with a particular focus on osteosarcoma. Furthermore, it explores the feasibility of UPR-targeted strategies as adjuvant or combinatorial approaches. In conclusion, this review provides an integrated and in-depth analysis on UPR-mediated mechanisms in sarcomas, offering perspectives on how targeting this pathway could accelerate the development of more effective and personalised treatments.” (page 1 and 2).

Comments 6: Introduction:

-          Consider adding one sentence with epidemiology/incidence to frame rarity and burden. 

Response 6: As suggested by the reviewer, we added a reference to sarcomas epidemiology. The text now states:  “…constituting 1% of all adult solid tumors…” (page 2).

Comments 7: -          End the section with an explicit aims paragraph (what the review covers, why the emphasis on OS, what is outside scope). 

Response 7: As suggested by the reviewer, we added such section. The text now states:

“This review explores current insights into UPR signaling in sarcomas, with a particular emphasis on osteosarcoma (OS), the subtype in which the role of the UPR has been most extensively studied. A detailed discussion of other stress-related pathways beyond the canonical UPR, or of endoplasmic reticulum (ER) stress responses independent of UPR activation, is beyond the scope of this work. Instead, we aim to highlight potential vulnerabilities and discuss the translational relevance of UPR targeting in these tumors.  By integrating molecular data, preclinical findings, and emerging clinical perspectives, this review seek to define the role of the UPR in sarcoma pathophysiology and its potential as a therapeutic avenue.” (page 3).

Comments 8: UPR biology (Figure 1):

-          The molecular overview is helpful. Please fix wording: “signal-transducting domain” → “signal-transducing domain.” 

Response 8: We thank the reviewer for their appreciation. We have fixed the misspelling (page 3).

Comments 9: -          Ensure the figure legend defines all acronyms introduced here (ATF6(N), ERAD, etc.) and cross-reference to the Abbreviations section.

Response 9: We improved the figure legend by adding a comprehensive description of the pathway and by adding all acronyms. The text now states: “Schematic representation of the Unfolded Protein Response (UPR) signaling. Unfolded or mis-folded proteins in the endoplasmic reticulum (ER) activate the three branches of the UPR: ATF6 (activating transcription factor 6), PERK (double-stranded RNA-activated protein kinase (PKR)–like ER kinase) and IRE1 (inositol-requiring enzyme 1). ATF6 is activated when BiP (binding im-munoglobulin protein) dissociates from its ER luminal domain and is transported to the Golgi for proteolytic cleavage by S1P (site-1 protease) and S2P (site-2 protease), which generates ATF6(N) (ATF6 N-terminal fragment). PERK is activated either by BiP dissociation or by direct binding of unfolded proteins. It phosphorylates eIF2α (α-subunit of eukaryotic initiation factor 2), which reduces protein translation and activates ATF4 (activating transcription factor 4). GADD34 (growth arrest and DNA damage–inducible gene 34) and PP1c (protein phosphatase 1) form a complex to dephosphorylate eIF2α. IRE1, activated by unfolded proteins or by BiP dissociation, cleaves XBP1 (X-box binding protein 1) mRNA producing XBP1s (XBP1 spliced). It also mediates the selective breakdown of mRNAs and miRNAs via regulated IRE1-dependent decay (RIDD). ATF6(N), ATF4 and XBP1s translocate to the nucleus, where they trigger the expression of target genes.” (page 3).

Comments 10: -          Minor style: consistently format “bZIP” (currently appears as “b-ZIP”). 

Response 10: We corrected the format accordingly.

Comments 11: UPR in cancer (angiogenesis, immune evasion, TME):

-          Consider adding a schematic that links UPR branches → downstream effectors → phenotypes (angiogenesis, MDSCs, T-cell dysfunction) to aid non-specialist readers.

Response 11: As suggested by the reviewer, we added a new figure (now figure 2 – page 8) for the section. The Figure provides a comprehensive summary of the molecular pathway activated by UPR in cancer cells.

Comments 12: UPR in sarcomas:

  1. Bone sarcomas — Osteosarcoma:

- Typos/clarity: “transforming growth factor beta (TFG-)” should be TGF-β; please correct throughout.

Response 12: We corrected the typos.

Comments 13: -          I suggest adding a summary table for OS (marker, model/patient cohort, readout, clinical correlation, n).

Response 13: As suggested by the reviewer, we added the OS summary table (now Table 1 – page 10 and 11).

Comments 14: 

Ewing’s sarcoma: -          Consider briefly noting whether EWS/FLI1 mechanistically engages UPR sensors directly/indirectly (as a hypothesis for future work).

Response 14: We added the insight as suggested by the reviewer. The paragraph now states:  

“Future studies should explore the mechanisms behind XBP1 activation. Special focus should be given to the regulatory role of the EWS/FLI1 fusion gene, which is central to this cancer, since current data reveal a major gap in our understanding.” (page 12).

Comments 15: Soft-tissue sarcomas (RMS, angiosarcoma, Kaposi’s)

-          RMS: Suggest adding exposure/dose context and whether effects persisted in 3D or in vivo models if available.

Response 15: As suggested by the reviewer, we added the concentrations of the drugs. The paragraph now states:  “… used at 10μM, 20μM and 40μM … used at 2μM, 5μM and 10μM … (MKC8866 used at 20μM and AMGEN44 used at 2μM)” and a comment about 3D and in vivo models “Further confirmation of these results in physiologically relevant 3D and in vivo models will be essential to establish their translational significance.” (page 13).

Comments 16: - Consider clarifying whether reduced IRE1α protein but retained kinase activity implies selective axis engagement in endothelium.

Response 16: The cited reference for angiosarcoma states that lower levels of Ire1 protein do not move accordingly to the levels of activation of the kinase domain, thus leading to a downstream activation of Xbp-1, despite reduced expression of the protein. Unfortunately, the paper does not go into further details, leaving the question of whether Ire1’s role is confined to Xbp-1 activation or whether other kinase-dependent downstream targets might also be involved in the regulation of angiogenesis. For this reason, we believe that no other conclusions, other than a kinase-dependent pro angiogenic acitivity can be specified. We have therefore only added a conclusive sentence that states: “Consistently, pharmacological inhibition of IRE1α kinase suppresses in vitro angiogenesis” (page 13).

Comments 17: - Add terse notes on leiomyosarcoma, liposarcoma, synovial sarcoma, MPNST, chondrosarcoma (even to state negative/limited evidence).

Response 17: As suggested by the reviewer, we added the few available information about other sarcomas in the “UPR in sarcomas” section. The added paragraph now states:

“Moreover, malignant peripheral nerve sheath tumors (MPNSTs) display elevated UPR markers, including BiP, eIF2α and XBP1s, compared with normal nerve tissue.” (page 14),

“In other sarcoma subtypes, like leiomyosarcoma, liposarcoma, synovial sarcoma and chondrosarcoma, the role of the UPR remains unexplored. Notably, in chondrosarcoma, only a single study reports that PRP-1 (proline-rich polypeptide), a toll-like receptor ligand, increases the expression of PERK, eIF2α, ATF4, CHOP, ATF6, IRE1α, and XBP1[124]; however, no studies have yet investigated the functional role of the UPR in this tumor.” and in the “targeting the UPR in cancer” section “In MPNST, BTZ-driven UPR activation enhances the synergistic antitumor efficacy observed when combinated with oHSV (oncolytic herpes simplex virus-1), while ER‑stress‑inducing agents such as thapsigargin, tunicamycin or the HSP90 inhibitor IPI‑504 cause UPR overexpression leading to cell death.” (page 14).

Comments 18: Targeting the UPR in cancer (and Table 1)

-          The preclinical compendium is useful and shows dual strategies (inhibit adaptive UPR vs push terminal UPR).

Response 18: We thank the reviewer for the appreciation.

Comments 19: Critical table issues to fix:

-          ORIN1001 vs MKC8866: The table labels “ORIN 1001 (MKC8866),” suggesting they are the same drug; they are distinct IRE1 RNase inhibitors and should not be conflated. Please correct drug names/rows and citations.

Response 19: To our knowledge, ORIN1001 and MKC8866 are actually the same drug (doi: 10.1016/j.isci.2023.106687 – “MKC8866 (recently renamed ORIN1001)”). We have therefore left the nomenclature unchanged.

Comments 20: Conclusions and Future Perspectives:

- Consider a closing research roadmap: (i) harmonized biomarker panels (ATF6α(N)/BiP/XBP1s), (ii) subtype- specific functional studies beyond OS, (iii) combination hypotheses (UPR modulators + DOX/BTZ/IO) with patient stratification by UPR activity.

Response 20: As suggested by the reviewer, we added such insights in the conclusions. The conclusions paragraph now states: “…integrating UPR-targeted approaches with standard-of-care treatments, like DOX and cis-platin,…” and “Moreover, comprehensive investigations are required to elucidate the functional relevance of the UPR within specific sarcoma subtypes. Future studies should begin by systematically characterizing standardized panels of UPR biomarkers, including IRE1, ATF6, PERK, BiP, and XBP1, to define their expression patterns and activation states across different sarcoma models. Subsequent efforts should aim to clarify how these signaling components contribute to distinct aspects of tumor biology, such as proliferation, survival, invasion, and therapeutic resistance, and to uncover the molecular pathways mediating these effects. Ultimately, in vivo studies will be essential to validate these findings and to assess the therapeutic potential of combinational strategies that target the UPR alongside conventional or novel anticancer treatments.” (page 16 and 17).

Reviewer 3 Report

Comments and Suggestions for Authors

The authors have written a review article encompassing current knowledge about the role of the UPR in sarcomas, with particular attention to osteosarcoma. The article is very well written. The introduction is well documented and sufficiently enriched with appropriate references. The subsections are also very well divided with appropriate contents in them which are very much informative. However, I have the following comments.
1. Please mention the search engines used to select the article for writing this review article.
2. In section 4.1.1. UPR in Osteosarcoma, third paragraph.

"......stream targets are enriched in OS cells and correlate with WNT and transforming growth factor beta (TFG-pathways, both known to promote OS progression...". There is a gap after the term TFG. Is it a typographical error or some term is missing; the first bracket also does not close. Please rectify.

3. There are many paragraphs which are small. They can be collated with the larger ones to make an appropriate sized paragraphs.
4. At the end, targeting of UPR section is present. In my opinion, this section can be expanded a little more. The role of nanostructures or nanoformulated drugs in intercepting UPRs can be included briefly in one paragraph. Many recent articles are available regarding this topic. https://doi.org/10.1016/j.biomaterials.2020.120452; https://doi.org/10.1038/s41416-023-02225-x.
5. The number of Figure are less. I recommend to include one more high resolution figure in the revised manuscript.
I recommend a minor revision.

Author Response

Comments 1: The authors have written a review article encompassing current knowledge about the role of the UPR in sarcomas, with particular attention to osteosarcoma. The article is very well written. The introduction is well documented and sufficiently enriched with appropriate references. The subsections are also very well divided with appropriate contents in them which are very much informative. However, I have the following comments.

Response 1: We thank the reviewer for the appreciation and for the constructive comments.

Comments 2: Please mention the search engines used to select the article for writing this review article.

Response 2: As suggested by the reviewer, we added the search engines used in the abstract: The text now states:  “Drawing on the current literature encompassing preclinical models, recent translational research (PubMed from 2000 to 2025),  and registered clinical trials (clinicaltrials.gov), this narrative review synthesizes current knowledge…” (page 1).

Comments 3: In section 4.1.1. UPR in Osteosarcoma, third paragraph.

"......stream targets are enriched in OS cells and correlate with WNT and transforming growth factor beta (TFG-pathways, both known to promote OS progression...". There is a gap after the term TFG. Is it a typographical error or some term is missing; the first bracket also does not close. Please rectify.

Response 3: We have corrected the typos.

Comments 4: There are many paragraphs which are small. They can be collated with the larger ones to make an appropriate sized paragraphs.

Response 4: We reduced the number of small paragraph throughout the review.

Comments 5: At the end, targeting of UPR section is present. In my opinion, this section can be expanded a little more. The role of nanostructures or nanoformulated drugs in intercepting UPRs can be included briefly in one paragraph. Many recent articles are available regarding this topic. https://doi.org/10.1016/j.biomaterials.2020.120452; https://doi.org/10.1038/s41416-023-02225-x.

Response 5: We thank the reviewer for the suggestion; we have now added a paragraph about nanostructures. The text now states:

“Recently, nanostructures and nanoformulated drugs have emerged as promising tools to modulate or intercept the UPR in cancer cells. In glioblastoma, the PDI inhibitor CCF642, especially when delivered via albumin-based nanoparticles, was shown to activate apoptosis-inducing UPR, downregulate PERK signaling, and restore sensitivity to temozolomide[143]. In lung‑carcinoma cells, the UPR is triggered by denatured proteins in the nanoparticle corona, which recruits the chaperone Hsp90ab1 to the particle surface and up‑regulates downstream pathways that promote EMT. Blocking Hsp90ab1 with geldanamycin prevents UPR-induced activation and EMT, pointing to the UPR-Hsp90ab1 axis as a driver of tumor cell reprogramming [144].” (page 15).

To expand the paragraph further, we have also added a brief description on what is known throughout literature on malignant peripheral nerve sheath tumor, the only other type of sarcoma with existing data. The text now states: “In MPNST, BTZ-driven UPR activation enhances the synergistic antitumor efficacy observed when combinated with oHSV (oncolytic herpes simplex virus-1) [141], while ER‑stress‑inducing agents such as thapsigargin, tunicamycin or the HSP90 inhibitor IPI‑504 cause UPR overexpression leading to cell death [123].“ (page 14 and 15).

Comments 6: The number of Figure are less. I recommend to include one more high resolution figure in the revised manuscript.

Response 6: As suggested by the reviewer, we added a new figure (now figure 2 – page 8) for a comprehensive description of the molecular pathways activated by the UPR.

Round 2

Reviewer 2 Report

Comments and Suggestions for Authors

The authors have adequately revised the manuscript. I recommend acceptance of the manuscript.